

# Pursuing the quest for better understanding the taxonomic distribution of the system of doubly uniparental inheritance of mtDNA

Arthur Gusman[1], Sophia Lecomte[2], Donald T. Stewart[3], Marco Passamonti[4] and Sophie Breton[1]

[1] Department of Biological Sciences, Université de Montréal, Montréal, Québec, Canada
[2] Department of Biological Sciences, Université de Strasbourg, Strasbourg, France
[3] Department of Biology, Acadia University, Wolfville, Nova Scotia, Canada
[4] Department of Biological Geological and Environmental Sciences, University of Bologna, Bologna, Italy

## ABSTRACT

There is only one exception to strict maternal inheritance of mitochondrial DNA (mtDNA) in the animal kingdom: a system named doubly uniparental inheritance (DUI), which is found in several bivalve species. Why and how such a radically different system of mitochondrial transmission evolved in bivalve remains obscure. Obtaining a more complete taxonomic distribution of DUI in the Bivalvia may help to better understand its origin and function. In this study we provide evidence for the presence of sex-linked heteroplasmy (thus the possible presence of DUI) in two bivalve species, i.e., the nuculanoid *Yoldia hyperborea* (Gould, 1841) and the veneroid *Scrobicularia plana* (Da Costa, 1778), increasing the number of families in which DUI has been found by two. An update on the taxonomic distribution of DUI in the Bivalvia is also presented.

## INTRODUCTION

Strict maternal inheritance (SMI) is considered to be the paradigm for mitochondrial DNA (mtDNA) transmission in animal species (*Birky, 2001*). One exception is found in bivalve molluscs, which possess a unique mode of mtDNA transmission named doubly uniparental inheritance (DUI) (*Hoeh, Blakley & Brown, 1991*; *Skibinski, Gallagher & Beynon, 1994*; *Zouros et al., 1994a*; *Zouros et al., 1994b*). DUI is characterized by the presence of two distinct sex-associated mitochondrial lineages: the female type (F mtDNA), which is transmitted through the eggs to all offspring, and the male type (M mtDNA) which is present in sperm, enters all eggs at the time of fertilization, but is only retained and transmitted through male offspring. In adults, the F-type mtDNA is predominant in all tissues of both sexes, except in the male gonad where the M-type mtDNA prevails; although some exceptions have been documented, adult females are essentially homoplasmic and adult males are heteroplasmic (reviewed in *Breton et al., 2007*; *Passamonti & Ghiselli, 2009*; *Zouros, 2013*). The stability of this system of heredity across evolutionary time in several

Corresponding author
Sophie Breton,
s.breton@umontreal.ca

orders of bivalves has produced highly divergent F and M mtDNAs: the mean nucleotide difference between both genomes is around 20% in many marine taxa (orders Mytiloida and Veneroida) and can reach >50% in freshwater mussels (order Unionoida) (*Breton et al., 2007*; *Doucet-Beaupré et al., 2010*). Although some major features of DUI are quite well known—for example species with DUI show strong sex biases in offspring towards one or the other sex following parental crosses (e.g., *Kenchington et al., 2002*; *Kenchington et al., 2009*), both F and M lineages show rapid molecular evolution compared to other animals, the M mtDNA usually evolves faster than the F mtDNA, M mitochondria show sex-specific behavior in newly formed zygotes, and novel mtDNA-encoded protein-coding genes have been found in species with DUI (*Breton et al., 2007*; *Passamonti & Ghiselli, 2009*; *Zouros, 2013*; *Breton et al., 2014*)—the main function of this peculiar system of mtDNA transmission still remains undetermined. Sustained by the correlation between DUI and gonochorism (and the absence of DUI and hermaphroditism), one main hypothesis suggests a link between this model of heredity and the maintenance of separate sexes (*Breton et al., 2011a*).

During the last decade, DUI has been described as a phenomenon that occurs in approximately 40 bivalve species (e.g., *Walker et al., 2006*; *Theologidis et al., 2008*; *Doucet-Beaupré et al., 2010*; *Dégletagne, Abele & Held, 2016*). Considering the great deal of new literature on DUI that has been done in the last few years, a more accurate count of species with DUI is clearly needed. Moreover, with ∼25,000 species (*BivAToL: Assembling the Bivalve Tree of Life, 2007*, http://www.bivatol.org), DUI is likely very widespread in the Bivalvia and it might be found in other molluscan groups as well (e.g., *Parakatselaki, Saavedra & Ladoukakis, 2015*; *Gusman, Azuelos & Breton, In press*). A broad mitochondrial survey of bivalves and other mollusc species is crucial to gauge the prevalence of DUI across molluscs and to evaluate its origin. To date, the vast majority of species with DUI that have been reported belong to the freshwater bivalve order Unionoida (families Hyriidae, Margaritiferidae, Unionidae) mostly because the PCR-based method used to detect DUI in this group, which is based on amplifying the *cox2* extension specific to unionoid male mtDNAs (*Curole & Kocher, 2002*), is simple and effective (*Walker et al., 2006*). The other groups in which species with DUI have been found are the orders Mytiloida (family Mytilidae) (*Hoeh, Blakley & Brown, 1991*; *Skibinski, Gallagher & Beynon, 1994*; *Zouros et al., 1994a*; *Zouros et al., 1994b*; *Passamonti, 2007*), Veneroida (families Arcticidae, Donacidae, Mactridae, Solenidae, Veneridae) (*Theologidis et al., 2008*; *Plazzi, 2015*; *Dégletagne, Abele & Held, 2016*), and Nuculanoida (family Nuculanidae), an order belonging to the most basal protobranch bivalve lineage (*Boyle & Etter, 2013*). It is still unsettled whether DUI has a single origin followed by its loss in several bivalve lineages or whether it has multiple and independent origins (*Hoeh et al., 1996*; *Theologidis et al., 2008*; *Zouros, 2013*; *Milani et al., 2014*). To disentangle these two possibilities, we must expand taxonomic sampling in a comprehensive manner.

The detection of DUI can be made by illustrating the presence of heteroplasmy in a male individual, specifically by retrieving different mitochondrial haplotypes from the male gonad and somatic tissues (the haplotype from male somatic tissues should be identical or more similar to the haplotype observed in female gonad and somatic tissues) and/or

by comparing male and female gonad/gamete samples (male sequences should always cluster together and female sequences too). Such an approach has already been successfully implemented in several previous DUI studies (e.g., *Passamonti & Scali, 2001*; *Theologidis et al., 2008*; *Boyle & Etter, 2013*; *Plazzi, Cassano & Passamonti, 2015*; *Plazzi, 2015*; *Vargas et al., 2015*; *Dégletagne, Abele & Held, 2016*). In the present study, we use this approach to test for the presence of DUI in two bivalve species, i.e., the nuculanoid *Yoldia hyperborea* (Gould, 1841) and the veneroid *Scrobicularia plana* (Da Costa, 1778), and we observe sex-linked heteroplasmy (thus the possible presence of DUI) in both of them, increasing the number of families in which DUI has been found by two. An update on the taxonomic distribution of DUI in the Bivalvia is also presented.

## MATERIALS AND METHODS

### Specimen's collection

Mature specimens of *Yoldia hyperborea* (Gould, 1841) were collected in the Baffin Sea (76°20′50N, 77°35′86W) in August 2013. *Scrobicularia plana* (Da Costa, 1778) samples were directly sent from the French National Museum of National History to our laboratory. All specimens were conserved in 95% ethanol. To identify sex-biased heteroplasmy, each individual was sexed by inspecting the gonads under a light microscope (100X) for the presence of eggs or sperm, and only individuals unambiguously sexed were kept for the present study. Dissections were carried on each individual to obtain somatic tissues (i.e., gills) and female or male gonad for DNA extractions (see below). The number of specimens analyzed for both species include 7 males and 7 females.

### DNA extraction, polymerase chain reaction amplification and sequencing

Total genomic DNA was extracted separately from gonad tissue and from gill tissue with a Qiagen DNeasy Blood & Tissue Kit (QIAGEN Inc., Valencia, CA, USA) using the animal tissue protocol. The quality and quantity of DNA, respectively, were assessed by electrophoresis on 1% agarose gels and with a BioDrop µLITE spectrophotometer. Before PCR amplifications, all samples were treated using OneStep[TM] PCR Inhibitor Removal Kit (Zymo Research, Irvine, CA) according to the manufacturer's protocol. For both species, partial sequence amplification of cytochrome oxidase subunit 1 (*cox1*) and large subunit ribosomal RNA (*rrnL* or *16S*) were carried out in 50 µl volumes comprising 5.0 µl 10X Taq buffer, 1.0 µl dNTP mix (10mM), 2.0 µl of each forward and reverse primer (10 µM; LCO1490 and HCO2198 for *cox1* (*Folmer et al., 1994*), and 16Sar and 16Sbr for *rrnL* (*Palumbi et al., 1991*)), 0.25 µl Taq DNA Polymerase (5U/µl; Bio Basic Inc., Markham, Ontario), 2 µl of DNA extract (100 ng/ul), and ddH2O up to 50 µl. Reactions were performed on a TProfessional Basic Thermocycler with the following PCR amplification conditions: initial denaturation at 95 °C for 2 min, followed by 35 cycles of 95 °C for 20 sec, 44 °C for 40 sec for *cox1* and temperature gradient 40–60 °C for 40 sec for *rrnL*, and 72 °C for 40 sec, followed by a final extension step at 72 °C for 5 min. The universal primers 16Sar and 16Sbr failed to amplify the *rrnL* sequence in both *S. plana* and *Y. hyperborea*. Resulting PCR products for *cox1* were visualized on 1% agarose gels under UV light with SYBR green

dye (Life Technologies), and purified with the Qiagen QIAquick PCR Purification Kit according to the manufacturer's protocol. The purified PCR products were sequenced at the Genome Quebec Innovation Centre (McGill University), using the Applied Biosystem's 3730xl DNA Analyzer technology.

## DNA cloning and sequencing

Examination of chromatograms revealed the presence of multiple sequencing peaks only in male gonad tissues of *S. plana* and *Y. hyperborea*, suggesting co-amplification of two different mtDNA types. The amplified products of male gonads were thus cloned using the PGEM-T Easy vector (Promega, Madison, WI, USA) to confirm the presence of F and M genomes. Ten recombinant clones, for each species, were sent to the Genome Quebec Innovation Centre to be sequenced on both strands using the primers pUC20 (5′-GTTTTCCCAGTCACGAC-3′) and pUC2 (5′-GAGCGGATAACAATTTCAC-3′).

## Sequence analysis

*Cox1* sequences were edited and aligned using MEGA 6 (version6.06; *Tamura et al., 2013*). Amino acid sequences were deduced using the invertebrate mitochondrial genetic code. Calculations of nucleotide and amino acid *p*-distances were performed with MEGA 6 (with 1,000 bootstrap replicates) (version6.06; *Tamura et al., 2013*).

Following a similar approach than *Dégletagne, Abele & Held (2016)* to look for the presence of two intraspecific "F and M" clades in *S. plana* and *Y. hyperborea*, maximum likelihood (using RAxML version 8.2.8; Stamakis, 2014) with bootstrap analyses (1,000 replicates) and Bayesian phylogenies (using MrBayes v3.2.6; *Huelsenbeck & Ronquist, 2001*; *Huelsenbeck & Ronquist, 2005*; *Ronquist et al., 2012*) were performed on *cox1* nucleotide sequences of both species with *Soletellina virescens* (Bivalvia, Veneridae, Genbank accession number: JN859944) and *Yoldia eightsii* (Bivalvia, Nuculanida, Genbank accession number: KJ571167) as outgroups for *S. plana* and *Y. hyperborea*, respectively (i.e., closest sequences according to BLAST search). Bayesian Information Criterion (BIC) (*Schwarz, 1978*) implemented in PartitionFinder (v1.1.1; *Lanfear et al., 2012*) was used to estimates the best-fitting models of evolution. Figtree (v1.4.2; *Morariu et al., 2008*) was used to edit the phylogenetic trees.

*S. plana* and *Y hyperborea* were included in an expanded phylogenetic analyses designed to verify molecular relationships among DUI species in general. Maximum likelihood (ML), maximum parsimony (MP) and Bayesian phylogenies were thus performed on F and M *cox1* nucleotide and amino acid sequences from all DUI species known to date and *Octopus vulgaris* (Mollusca: Octopoda) and *Aplysia californica* (Mollusca: Gastropoda) were used as outgroup taxa. F and M *cox1* sequences other than those obtained in the present study for *S. plana and Y. hyperborea* were retrieved from Genbank; the complete phylogenetic dataset is shown in Table 1. *Cox1* sequences were aligned using MEGA 6 (version6.06; *Tamura et al., 2013*) and the best-fitting models of DNA evolution were selected using PartitionFinder (v1.1.1; *Lanfear et al., 2012*) according to BIC values (*Schwartz,1978*). Best models were applied whenever possible. Data were partitioned according to nucleotide position and gaps were treated as missing data.

**Table 1  Complete phylogenetic dataset used for phylogenetic reconstruction.** GenBank accession numbers of sequences are listed in the last two columns. Sequences obtained for the present study are indicated in bold.

| Species | Authority | M *cox1* | F *cox1* |
| --- | --- | --- | --- |
| *Actinonaias ligamentina* | (Lamarck, 1819) | AF406796 | AF231730 |
| *Amblema plicata* | (Say, 1817) | EF033295 | EF033258 |
| *Anodonta californiensis* | (Lea, 1852) | AY493507 | AY493462 |
| *Anodonta oregonensis* | (Lea, 1838) | AY493504 | AY493480 |
| *Anodonta wahlamatensis* | (Lea, 1838) | AY493493 | AY493467 |
| *Anodonta woodiana* | (Lea, 1834) | EF440350 | HQ283346 |
| *Aplysia californica* | (Cooper, 1863) | N/A | NC005827 |
| *Brachidontes exustus* | (Linnaeus, 1758) | AY621946 | NC024882 |
| *Brachidontes pharaonis* | (Fischer, 1870) | DQ836012 | DQ836013 |
| *Brachidontes variabilis* | (Krauss 1848) | DQ836020 | DQ836019 |
| *Cumberlandia monodonta* | (Say, 1829) | AY785397 | KF647529 |
| *Cyrtonaias tampicoensis* | (Lea, 1838) | EF033299 | EF033259 |
| *Fusconaia flava* | (Rafinesque, 1820) | EF033307 | EF033261 |
| *Glebula rotundata* | (Lamarck, 1819) | EF033304 | EF033264 |
| *Graptacme eborea* | (Conrad, 1846) | N/A | AY260825 |
| *Hamiota subangulata* | (Lea, 1840) | EF033305 | EF033266 |
| *Echyridella menziesii* | (Gray, 1843) | AF406802 | AF231747 |
| *Hyriopsis cumingii* | (Lea, 1852) | KC150028 | HM347668 |
| *Unio japanensis* | (Lea, 1859) | AB055624 | AB055625 |
| *Lamprotula leai* | (Griffith, 1833) | KC847114 | JQ691662 |
| *Lamprotula tortuosa* | (Lea, 1865) | KC441487 | KC109779 |
| *Lampsilis hydiana* | (Lea, 1838) | EF033298 | EF033270 |
| *Lampsilis ovata* | (Say, 1817) | EF033303 | EF033262 |
| *Lampsilis siliquoidea* | (Barnes, 1823) | KC408795 | KC408768 |
| *Lampsilis straminea* | (Conrad, 1834) | EF033297 | EF033271 |
| *Lampsilis teres* | (Rafinesque, 1820) | AF406794 | KT285644 |
| *Lemiox rimosus* | (Rafinesque, 1831) | EF033302 | EF033256 |
| *Ligumia recta* | (Lamarck, 1819) | AF406795 | KC291717 |
| *Margaritifera margaritifera* | (Linnaeus, 1758) | AY785399 | KC429108 |
| *Meretrix Lamarckii* | (Deshayes, 1853) | KP244452 | KP244451 |
| *Musculista senhousia* | (Benson, 1842) | AY570050 | AY570041 |
| *Mytella charuana* | (Soot-Ryen, 1955) | JQ685159 | JQ685156 |
| *Mytilus californianus* | (Conrad, 1837) | JX486123 | JX486124 |
| *Mytilus edulis* | (Linnaeus, 1758) | AY484747 | HM489873 |
| *Mytilus galloprovincialis* | (Lamarck, 1819) | AY363687 | AY497292 |
| *Mytilus trossulus* | (Gould, 1850) | GQ438250 | AY823625 |
| *Obliquaria reflexa* | (Rafinesque, 1820) | EF033292 | EF033254 |
| *Obovaria olivaria* | (Rafinesque, 1820) | EF033306 | EF033267 |
| *Octopus vulgaris* | (Cuvier, 1797) | N/A | AB191269 |

| Species | Authority | M *cox1* | F *cox1* |
|---|---|---|---|
| *Plectomerus dombeyanus* | (Valenciennes, 1827) | EF033290 | EF033252 |
| *Pleurobema sintoxia* | (Rafinesque, 1820) | EF033291 | EF033253 |
| *Popenaias popeii* | (Lea, 1857) | EF033294 | EF033257 |
| *Potamilus purpuratus* | (Lamarck, 1819) | AF406797 | AF406804 |
| *Pseudocardium sachalinense* | (Schrenck, 1862) | KJ650517 | KJ650515 |
| *Ptychobranchus fasciolaris* | (Rafinesque, 1820) | EF033301 | EF033265 |
| *Pyganodon fragilis* | (Lamarck, 1819) | AF406800 | AF406805 |
| *Pyganodon grandis* | (Say, 1829) | FJ809755 | FJ809754 |
| *Quadrula quadrula* | (Rafinesque, 1820) | FJ809751 | FJ809750 |
| *Quadrula refulgens* | (Lea, 1868) | EF033309 | EF033269 |
| *Scrobicularia plana* | (Da Costa, 1778) | **KX447424** | **KX447420** |
| *Solenaia carinatus* | (Heude, 1877) | KC848655 | KC848654 |
| *Toxolasma glans* | (Lea, 1840) | EF033293 | EF033255 |
| *Unio crassus* | (Philipson, 1788) | EU548052 | KJ525915 |
| *Unio pictorum* | (Linnaeus, 1758) | EU548055 | HM014133 |
| *Unio tumidus* | (Philipson, 1788) | EU548054 | KC703957 |
| *Utterbackia peninsularis* | (Bogan & Hoeh, 1995) | HM856635 | HM856636 |
| *Venerupis philippinarum* | (Adams, 1850) | AB065374 | AB065375 |
| *Venustaconcha ellipsiformis* | (Conrad, 1836) | FJ809752 | FJ809753 |
| *Yoldia hyperborea* | (Gould, 1841) | **KX447428** | **KX447425** |

ML analyses were conducted with RAxML (version 8.2.8; *Stamatakis, 2014*). A non-parametric bootstrap (*Felsenstein, 1985*) analysis was performed, using 1,000 bootstrap replicates and 20 ML searches, to assess nodal support for both trees. Outgroups were set to be paraphyletic to the monophyletic ingroup. MP analyses were carried out using PAUP software (v 4.0a147; *Swofford, 2001*). To optimize the chance of having the best topology, 100 random stepwise additions under tree-bisection reconnection branch swapping were applied (*Bogan & Hoeh, 2000*). Reliability of the internal nodes was evaluated by 1,000 pseudoreplicates using the heuristic search algorithm. Bayesian analyses were conducted using MrBayes (v3.2.6; *Huelsenbeck & Ronquist, 2001*; *Huelsenbeck & Ronquist, 2005*; *Ronquist et al., 2012*). Each analysis consisted of two independent runs of 4 MC$^3$ chains that were run for 10,000,000 generations. Convergence was estimated through the log likelihood value of trees, potential scale reduction factor (PSRF) and standard deviation of average split frequencies sampled every 1,000 generations (*Gelman & Rubin, 1992*). Trees were sampled every 100 generations and a majority-rule consensus tree was computed after discarding the first 25% as burn-in. Fidelity of the topology was evaluated with the posterior probabilities from the consensus tree. All phylogenetic trees were edited for easier readability using FigTree (v1.4.2; *Morariu et al., 2008*).

All the alignments used for phylogenetic reconstruction are available here (https://dx.doi.org/10.6084/m9.figshare.3798789.v1).
**Table 2** **Number of sequences obtained for *S. plana* and *Y. hyperborea*.** The number of haplotypes indicated regroups both F and M sequences. Genbank accession numbers are listed. Gi, Gills; Go, Gonads.

| Species | *Cox1* sequences | | *Cox1* haplotypes | |
|---|---|---|---|---|
| | Female Gi/Go | Male Gi/Go | Count | Genbank ids |
| *Scrobicularia plana* | 3/3 | 5/6 | 5 | KX447420, KX447421, KX447422, KX447423, KX447424 |
| *Yoldia hyperborea* | 7/4 | 7/7 | 4 | KX447425, KX447426, KX447427, KX447428 |

**Table 3** **Nucleotide and amino acid Pairwise-distance for *S. plana* and *Y. hyperborea cox1* sequences.** Standard errors are given under p-distance values. Bold numbers indicate significant values for the presence of DUI. F, intrafemale divergence; M, intramale divergence; F/M divergence between males and females.

| Species | *cox1* nucleotide sequences | | | *cox1* amino acid sequences | | |
|---|---|---|---|---|---|---|
| | F | M | F/M | F | M | F/M |
| *Scrobicularia plana* | 0.0067 | 0.0020 | **0.0965** | 0.0072 | 0.0000 | **0.0659** |
| | ±0.0034 | ±0.0014 | ±0.0074 | ±0.0070 | ±0.0000 | ±0.0100 |
| *Yoldia hyperborea* | 0.0014 | 0.0000 | **0.0596** | 0.0000 | 0.0000 | **0.0454** |
| | ±0.0008 | ±0.0000 | ±0.0079 | ±0.0000 | ±0.0000 | ±0.0124 |

## RESULTS

### Genetic distances

For this study, two new species were tested for the presence of DUI and 42 sequences were examined in total: the number of sequences and haplotypes for each species are listed in Table 2. Different haplotypes (i.e., for the same tissue among females or males, and thus non related to DUI) were found for both species (Table 2). All mtDNA sequences are available via GenBank under accession numbers KX447420, KX447421, KX447422, KX447423, KX447424, KX447425, KX447426, KX447427, KX447428. Sequences with the same haplotype were deposited only once.

Intragroup (female sequences and male sequences, respectively) and intergroup (female versus male sequences) nucleotide and amino acid *p*-distances are shown for both species in Table 3. The *p*-distances between female and male sequences within each species are significantly larger than the within group *p*-distances. Specifically, for *Y. hyperborea cox1* sequences, intragroup *p*-distances are low, i.e., 0.0014 for female sequences and zero for male sequences (i.e., male sequences are identical), with standard error of ±0.0008 for female sequences, whereas the between group *p*-distance is considerably higher with a value of 0.0596 ± 0.0079 (amino acid *p*-distance is 0.0454 ± 0.0124). The same observation can be made for *S. plana cox1* sequences: within group *p*-distances are 0.0067 ± 0.0034 and 0.0020 ± 0.0014, for female and male sequences, respectively, whereas the between group *p*-distance value is 0.0965 ± 0.0074 (amino acid *p*-distance is 0.0659 ± 0.0100).

## Phylogenetic analyses for Scrobicularia plana and Yoldia hyperborea

Phylogenetic analyses of partially sequenced *S. plana cox1* and *Y. hyperborea cox1* sequences were conducted using the HKY + G model (*Hasegawa, Kishino & Yano, 1985*). Based on high bootstraps and posterior probability values, female and male haplotypes were clustered into two well-supported clades for each species (Figs. S1 and S2).

## Taxonomic distribution of DUI

Genetic distances and phylogenetic analyses provide evidence for sex-associated mitochondrial heteroplasmy in *Scrobicularia plana* (Bivalvia: Semelidae) and *Yoldia hyperborea* (Bivalvia: Yoldiidae), raising the total of bivalve families in which DUI has been discovered to 12 (*Hoeh, Stewart & Guttman, 2002*; *Theologidis et al., 2008*; *Boyle & Etter, 2013*; *Plazzi, 2015*; *Dégletagne, Abele & Held, 2016*).

The list of the 103 bivalve species in which DUI has been found to date is presented in Table 4. Evidence for DUI is reported in the literature for 92 species (and for two species in the present study). For the other nine species, the evidence is based on sequences derived from male and female gonads and retrieved from GenBank (see Tables 1 and 4). These sequences all show strong nucleotide divergence between mitochondrial gene sequences obtained from male and female individuals, with F to M type *p*-distances ranging from 10% to 30%.

## Phylogenetic analyses

A total of 608 *cox1* nucleotide sequences were aligned for a total of 114 operational taxonomic units (OTUs) for phylogenetic analyses. GTR + I + G (*Tavaré, 1986*) was selected as the best-fitting model of evolution for nucleotides (Table S1). The majority rule nucleotide-based BI tree was favored as our best tree obtained for this study (Figs. 1–4). It shows strong topology similarity with both MP and ML trees (Figs. S3 and S4).

Main features of the BI tree (Figs. 1–4) are as follows: (1) pteriomorph (Mytiloida) + *Yoldia hyperborea* (PP = 0,80) and heterodont bivalves (Veneroida) (PP = 0,99) are reciprocally monophyletic with palaeoheterodont (Unionoida) bivalves being the sister group to these clades (PP = 0,86); (2) the order Veneroida is well resolved with 3 nodes (PP = 1.00) separating each superfamily: Veneroidea, Tellinoidea and Mactroidea; (3) in both F and M clusters of the order Unionoida, the superfamily Hyrioidea represented by *H. menziesi* is a positioned as a well separated sister group (PP = 1.00) to the Unionoidea superfamily (Note: within the Unionoidea, the separation between the two families Margaritiferidae and Unionidae is also apparent and well supported only in the M cluster (PP = 0.93)); (4) the F and M clades are reciprocally monophyletic only in unionoids and *Mytilus* spp., i.e., the F sequences of different species cluster together as do the M sequences, all the other species exhibit a phylogenetic pattern where F and M sequences clusters are distinct from one another but are nonetheless sister groups; and (5) branch lengths indicate a higher substitution rate for the M genomes relative to that of the F genomes for almost all species.

**Table 4 List of species with DUI known to date.** The taxonomic affiliation is made according to *Giribet & Wheeler (2002)*. Information about the presence of DUI was retrieved according to the references listed in the last column.

| Superfamily/family | Species | References |
|---|---|---|
| Unionoidea/Unionidae | *Actinonaias ligamentina* | *Hoeh, Stewart & Guttman (2002)* |
| | *Amblema plicata* | *Curole & Kocher (2005)* |
| | *Anodonta anatina* | *Soroka (2010)* |
| | *Anodonta californensis* | *Mock et al. (2004)* |
| | *Anodonta implicata* | *Curole & Kocher (2002)* |
| | *Anodonta oregonensis* | *Mock et al. (2004)* |
| | *Anodonta wahlamatensis* | *Mock et al. (2004)* |
| | *Anodonta woodiana* | *Soroka (2005)*; *Soroka (2008)* |
| | *Cyprogenia alberti* | *Walker et al. (2006)* |
| | *Cyrtonaias tampicoensis* | *Hoeh, Stewart & Guttman (2002)* |
| | *Dromus dromas* | *Walker et al. (2006)* |
| | *Ellipsaria lineolata* | *Walker et al. (2006)* |
| | *Elliptio dilitata* | *Walker et al. (2006)* |
| | *Epioblasma brevidens* | *Walker et al. (2006)* |
| | *Glebula rotundata* | *Curole & Kocher (2005)* |
| | *Gonidea angulata* | *Walker et al. (2006)* |
| | *Hamiota subangulata* | *Chapman et al. (2008)* |
| | *Hyriopsis cumingii* | KC471519 |
| | *Hyriopsis schlegelii* | HQ641407 |
| | *Inversidens japanensis* | *Doucet-Beaupré et al. (2010)* |
| | *Lamprotula leai* | KC847114 |
| | *Lamprotula tortuosa* | KC471516 |
| | *Potamilus purpuratus* | *Hoeh, Stewart & Guttman (2002)* |
| | *Lampsilis cardium* | *Walker et al. (2006)* |
| | *Lampsilis hydiana* | *Walker et al. (2006)* |
| | *Lampsilis ovata* | *Chapman et al. (2008)* |
| | *Lampsilis powellii* | *Walker et al. (2006)* |
| | *Lampsilis reeveiana* | *Walker et al. (2006)* |
| | *Lampsilis siliquoidea* | *Walker et al. (2006)* |
| | *Lampsilis streckeri* | *Walker et al. (2006)* |
| | *Lampsilis straminea* | *Curole & Kocher (2002)* |
| | *Lampsilis teres* | *Hoeh, Stewart & Guttman (2002)* |
| | *Lanceolaria grayana* | AB040829, AB040830 |
| | *Lasmigona complanata* | *Stewart et al. (2013)* |
| | *Lasmigona costata* | *Stewart et al. (2013)* |
| | *Lemiiox rimosus* | *Chapman et al. (2008)* |
| | *Leptodea fragilis* | *Walker et al. (2006)* |
| | *Leptodea leptodon* | *Walker et al. (2006)* |
| | *Ligumia recta* | *Hoeh, Stewart & Guttman (2002)* |
| | *Margaritifera marrianae* | *Stewart et al. (2013)* |

**Table 4** (*continued*)

| Superfamily/family | Species | References |
|---|---|---|
| | *Medionidus conradicus* | *Walker et al. (2006)* |
| | *Obliquaria reflexa* | *Chapman et al. (2008)* |
| | *Obovaria olivaria* | *Walker et al. (2006)* |
| | *Plectomerus dombeyanus* | *Curole & Kocher (2005)* |
| | *Pleurobema sintoxia* | *Chapman et al. (2008)* |
| | *Popenaias popeii* | *Walker et al. (2006)* |
| | *Potamilus alatus* | *Walker et al. (2006)* |
| | *Potamilus capax* | *Walker et al. (2006)* |
| | *Potamilus ohiensis* | *Walker et al. (2006)* |
| | *Pseudodon vondembuschianus* | *Walker et al. (2006)* |
| | *Ptychobranchus fasciolare* | *Walker et al. (2006)* |
| | *Pyganodon fragilis* | *Hoeh et al. (1996)* |
| | *Pyganodon grandis* | *Liu, Mitton & Wu (1996)* |
| | *Fusconaia flava* | *Hoeh et al. (1996)* |
| | *Quadrula quadrula* | *Curole & Kocher (2002)* |
| | *Quadrula refulgens* | *Curole & Kocher (2002)* |
| | *Solenaia carinatus* | *Huang et al. (2013)* |
| | *Strophitus undulatus* | *Stewart et al. (2013)* |
| | *Toxolasma glans* | *Stewart et al. (2013)* |
| | *Toxolasma lividus* | *Stewart et al. (2013)* |
| | *Toxolasma minor* | *Stewart et al. (2013)* |
| | *Toxolasma paulus* | *Stewart et al. (2013)* |
| | *Truncilla truncate* | *Walker et al. (2006)* |
| | *Unio crassus* | *Soroka (2010)* |
| | *Unio delphinus* | *Machordom et al. (2015)* |
| | *Unio pictorum* | *Soroka (2010)* |
| | *Unio tumidus* | *Soroka (2010)* |
| | *Utterbackia peggyae* | *Breton et al. (2011a)* |
| | *Utterbackia peninsularis* | *Breton et al. (2011a)* |
| | *Venustaconcha ellipsiformis* | *Chakrabarti et al. (2006)* |
| | *Villosa lienosa* | *Curole & Kocher (2005)* |
| | *Villosa villosa* | *Walker et al. (2006)* |
| Unionoidea/Margaritiferidae | *Cumberlandia monodonta* | *Breton et al. (2011a)* |
| | *Dahurinaia dahurica* | *Walker et al. (2006)* |
| | *Margaritifera hembeli* | *Curole & Kocher (2005)* |
| | *Margaritifera margaritifera* | *Hoeh, Stewart & Guttman (2002)* |
| Hyrioidea/Hyriidae | *Hyridella menziesi* | *Hoeh, Stewart & Guttman (2002)* |

**Table 4** (*continued*)

| Superfamily/family | Species | References |
|---|---|---|
| Mytiloidea/Mytilidae | *Brachidontes exustus* | *Lee & Foighil (2004)* |
| | *Brachidontes pharaonis* | *Lee & Foighil (2004)* |
| | *Brachidontes variabilis* | *Terranova et al. (2007)* |
| | *Geukensia demissa* | *Hoeh et al. (1996)* |
| | *Modiolus modiolus* | *Robicheau et al. (in press)* |
| | *Musculista senhousia* | *Passamonti (2007)* |
| | *Mytella charuana* | *Alves et al. (2012)* |
| | *Mytilus californianus* | *Beagley, Taylor & Wolstenholme (1997)* |
| | *Mytilus coruscus* | *Breton et al. (2011b)* and AF315572 |
| | *Mytilus edulis* | *Zouros et al. (1994a)* |
| | *Mytilus galloprovincialis* | *Quesada, Skibinski & Skibinski (1996)* |
| | *Mytilus trossulus* | *Zouros et al. (1994a)* |
| | *Perumytilus purpuratus* | *Vargas et al. (2015)* |
| Arcticoidea/Arcticidae | *Arctica islandica* | *Dégletagne, Abele & Held (2016)* |
| Mactroidea/Mactridae | *Pseudocardium sachalinense* | *Plazzi (2015)* |
| Solenoidea/Solenidae | *Solen grandis* | AB064984, AB064985 |
| Tellinoidea/Donacidae | *Donax cuneatus* | *AB040841, AB040842* |
| | *Donax faba* | *AB040843, AB040844* |
| | *Donax trunculus* | *Theologidis et al. (2008)* |
| Tellinoidea/Semelidae | *Scrobicularia plana* | Present study |
| Veneroidea/Veneridae | *Cyclina sinensis* | *AB040833, AB040834* |
| | *Venerupis philippinarum* | *Passamonti & Scali (2001)* |
| | *Meretrix Lamarckii* | *Plazzi, Cassano & Passamonti (2015)* |
| Nuculanoidea/Nuculanidae | *Ledella sublevis* | *Boyle & Etter (2013)* |
| | *Ledella ultima* | *Boyle & Etter (2013)* |
| Nuculanoidea/Yoldiidae | *Yoldia hyperborea* | Present study |

# DISCUSSION

## Evidence for DUI in Scrobicularia plana and Yoldia hyperborea

Both *p*-distance and phylogenetic analyses indicate the coexistence of sex-linked F and M mitochondrial lineages in *S. plana* and *Y. hyperborea* (Figs. 1–4; Table 3 and Figs. S1 and S2). As mentioned earlier, the strategy of searching for sex-biased heteroplasmy of mitochondrial gene sequences as a means for detecting DUI has been successfully employed in other bivalve species (e.g., *Boyle & Etter, 2013*; *Vargas et al., 2015*; *Dégletagne, Abele & Held, 2016*). DUI can also be detected by *in vivo* localization of male mitochondria in embryos: an aggregate pattern of M-type mitochondria only into the male germline during early embryonic stages is typical of DUI species (*Cao, Kenchington & Zouros, 2004*; *Obata & Komaru, 2005*; *Cogswell, Kenchington & Zouros, 2006*; *Milani, Ghiselli & Passamonti, 2012*). The percentage of nucleotide divergence between the F and M *cox1* sequences for both of these species, i.e., 33.4% for *S. plana* and 13% for *Y. hyperborea*, is within the range of

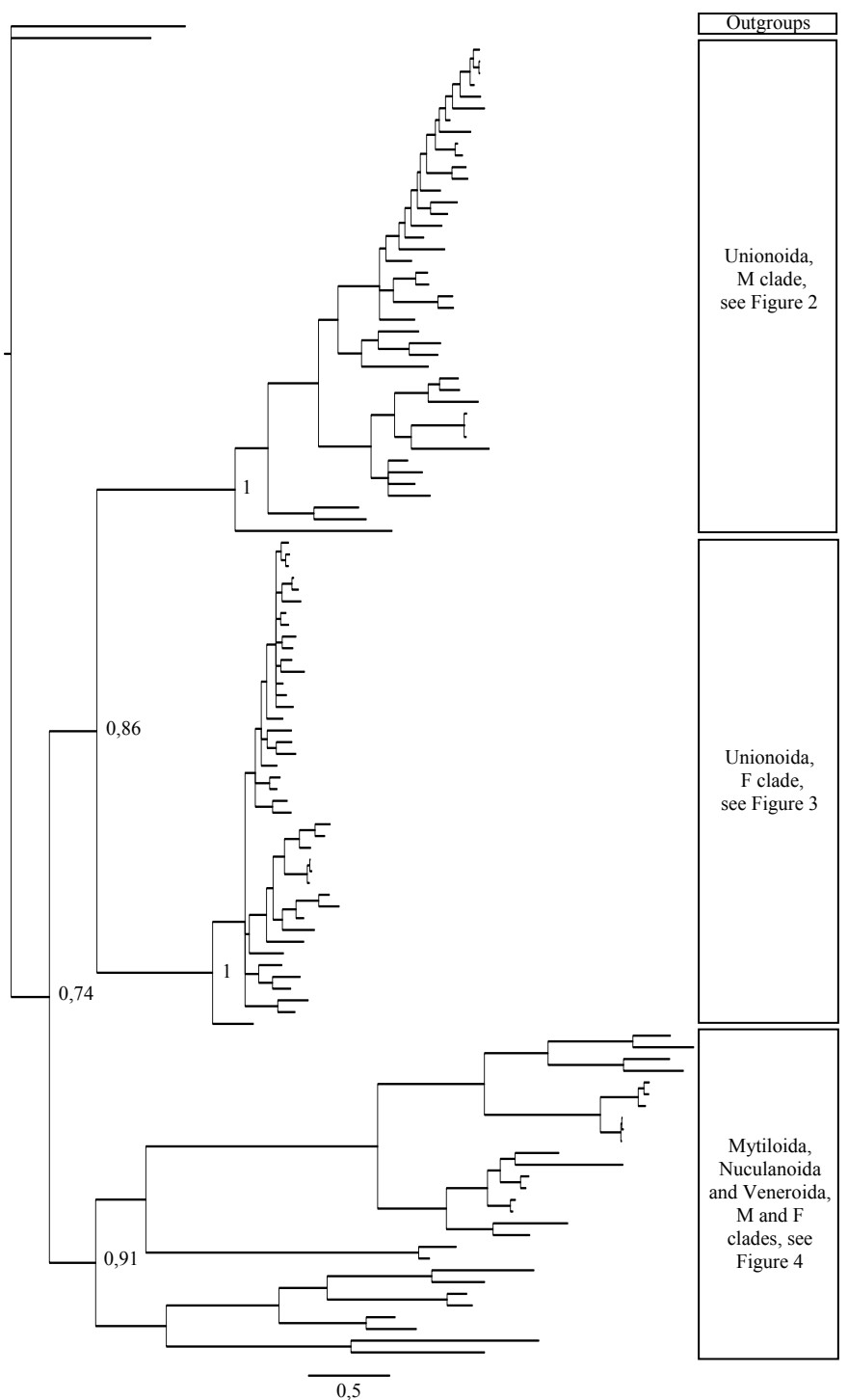

**Figure 1** **Bayesian inference majority-rule tree of bivalve *cox1* partial sequence.** Relationships based on an analysis using the GTR + I + G model. Numbers at each node indicate nodal support.

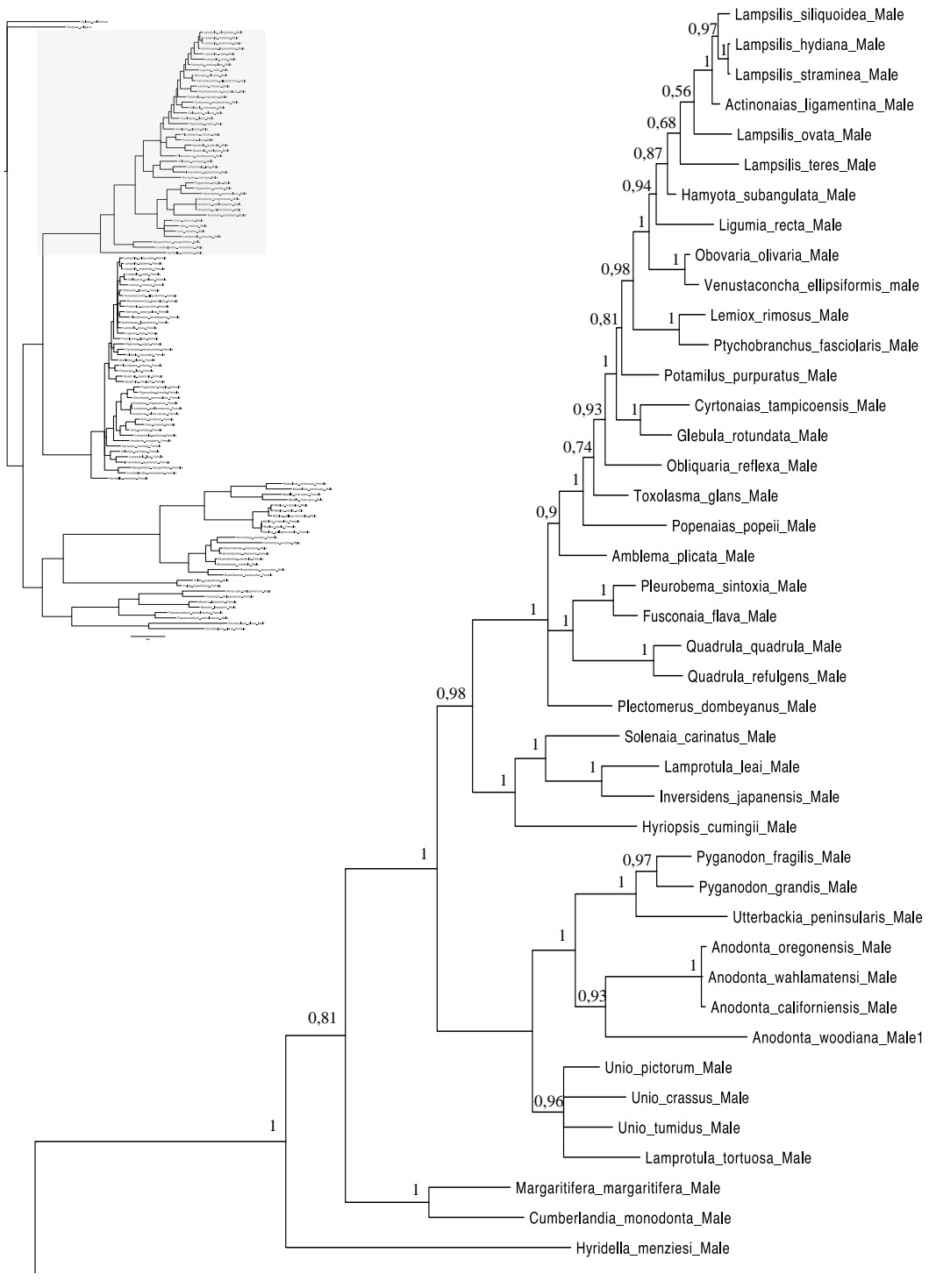

**Figure 2  Bayesian inference majority-rule tree of bivalve *cox1* partial sequence.**  Relationships based on an analysis using the GTR + I + G model. Unionoida M clade. Numbers at each node indicate nodal support.

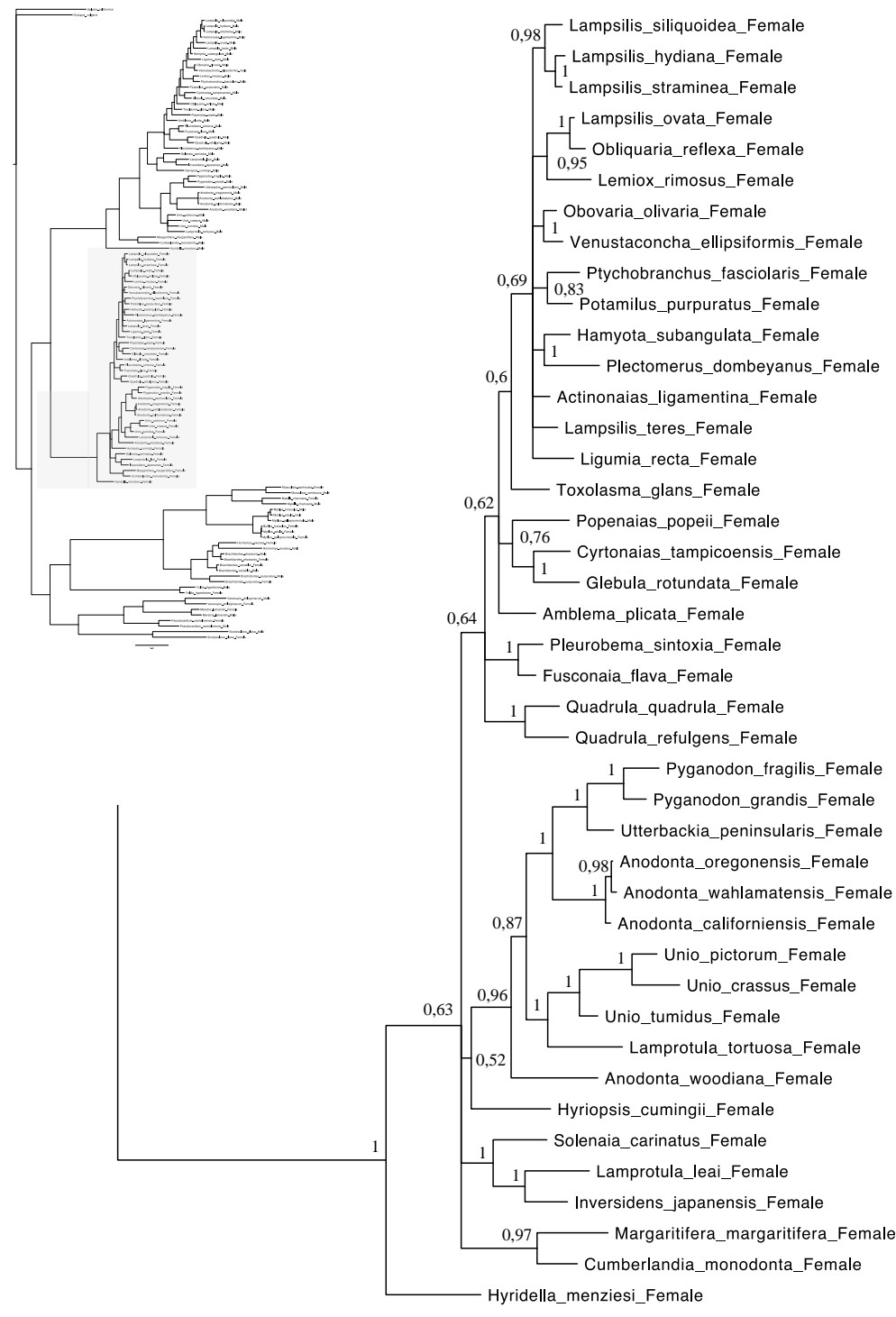

**Figure 3** **Bayesian inference majority-rule tree of bivalve *cox1* partial sequence.** Relationships based on an analysis using the GTR + I + G model. Unionoida F clade. Numbers at each node indicate nodal support.

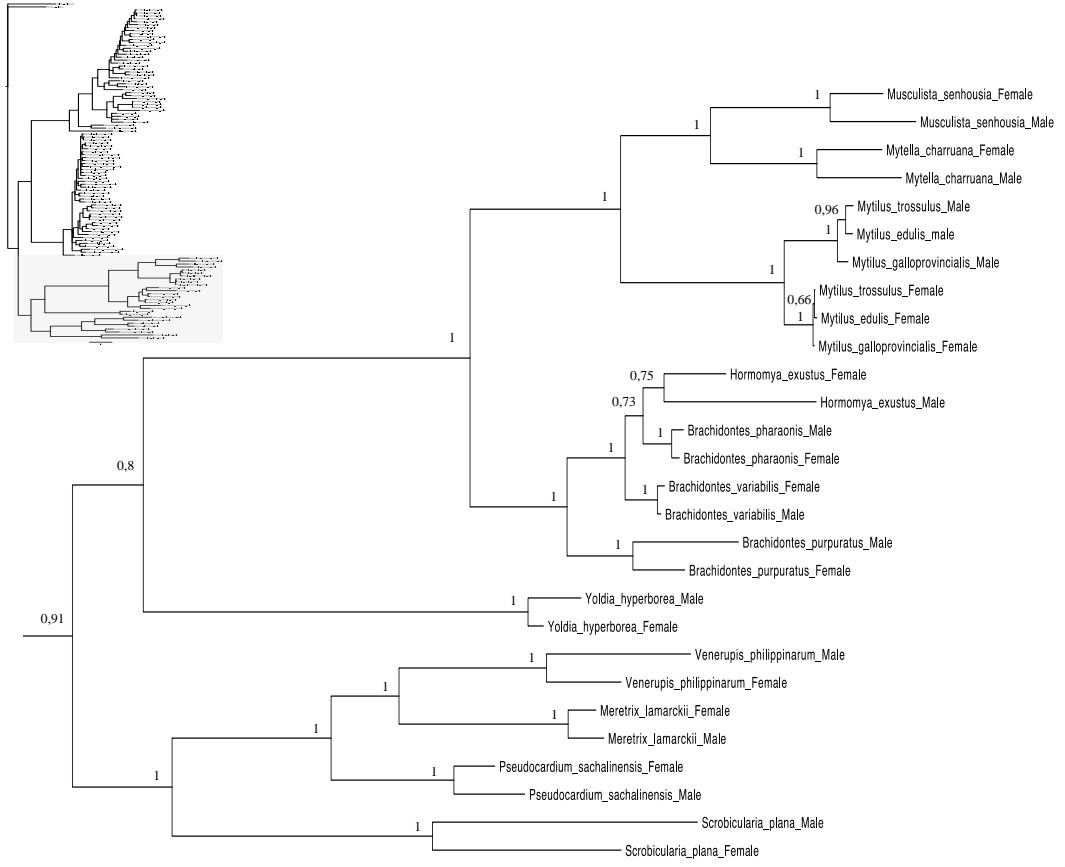

**Figure 4** **Bayesian inference majority-rule tree of bivalve *cox1* partial sequence.** Relationships based on an analysis using the GTR + I + G model. Mytiloida, Nuculanoida and Veneroida, M and F clades. Numbers at each node indicate nodal support.

what has been found for other bivalves with DUI (e.g., 8% in the veneroid *Artica islandica* (*Dégletagne, Abele & Held, 2016*), 17% in the nuculanoid *Ledella sublevis* (*Boyle & Etter, 2013*); 24% in the mytiloid *Mytilus edulis* (*Breton et al., 2006*),and 50% in the unionoid *Inversidens japanensis* (*Doucet-Beaupré et al., 2010*)).

The discovery of DUI in *Y. hyperborea* provides the first example of this unusual system of mitochondrial DNA transmission in the family Yoldiidae, and the third case for the order Nuculanoida (subclass Protobranchia; *Boyle & Etter, 2013*). Protobranchia being the most basal lineage within the Bivalvia (*Giribet & Wheeler, 2002*; *Smith et al., 2011*), this result suggests that the origin of DUI is older than the birth of the autolamellibranchiata (*Theologidis et al., 2008*; *Doucet-Beaupré et al., 2010*; *Boyle & Etter, 2013*). However, of the three protobranch orders, only the Nuculanoida, has been reported to exhibit DUI (*Boyle & Etter, 2013*), and the phylogenetic status of this order is still being questioned. Indeed, recent phylogenetic studies suggest that the Nuculanoida is not a member of the basal protobranch group, which includes Nuculoida and Solemyoida, but instead is associated with the Pteriomorpha (*Wilson, Rouse & Giribet, 2010* (based on 3 nuclear and

2 mitochondrial genes); *Plazzi et al., 2011*) (based on 4 mitochondrial genes); Breton et al., unpublished (based on complete mitochondrial genomes and 3 nuclear genes combined with an extensive morphological dataset)), whereas other recent studies based on four nuclear genes (*Sharma et al., 2012*) and phylogenomic data (*Smith et al., 2011*; *González et al., 2015*) supported the monophyly of Nuculanoida + Opponobranchia (Nuculoida and Solemyoida; *Giribet, 2008*). The presence of DUI in protobranchs thus remains an open question until the publication of a well-supported and robust phylogeny of bivalves showing the monophyly of the traditional clade Protobranchia (i.e., Solemyoida + Nuculoida + Nuculanoida) and/or until the discovery of DUI in nuculoid or solemyoid bivalves.

The peppery furrow shell *Scrobicularia plana* belongs to the order Veneroida. It is the first reported species with DUI from the family Semelidae, raising the total number of veneroid families in which DUI has been discovered to six (*Theologidis et al., 2008*; *Plazzi, 2015*; *Dégletagne, Abele & Held, 2016*; present study). Apart from providing new insights into the taxonomic distribution of DUI, *S. plana* may play a key role for better understanding the hypothesized role of DUI in sex determination (e.g., *Breton et al., 2007*; *Breton et al., 2011a*; *Breton et al., 2014*; *Breton & Stewart, 2015*; *Mitchell et al., 2016*). Indeed, an "intersex" condition, i.e., the appearance of oocytes in male gonads following endocrine disruption, has been reported in *S. plana* and is associated with differentially expressed mitochondrial transcripts in males exhibiting intersex compared to "normal" males (*Chesman & Langston, 2006*). Specifically, using a suppressive subtractive hybridization approach, *Ciocan et al. (2012)* were able to determine that several mitochondrial mRNA transcripts were down-regulated in clam intersex samples (i.e., *cox1, cytb, nad1, nad2, nad3, nad4*). Interestingly, we observed that the down-regulated *cox1* sequence identified by *Ciocan et al. (2012)* was identical to the male *cox1* sequences from our study, indicating that the down-regulation of male mitochondrial sequences is associated with the appearance of female characteristics in male gonads in this species. These results provide more evidence for a link between DUI and sex determination or differentiation. It is noteworthy that the intersex has been shown to be a widespread phenomenon in bivalves, including in species with DUI (e.g., *R. philippinarum* (*Lee et al., 2010*) and *M. galloprovincialis* (*Ortiz-Zarragoitia & Cajaraville, 2010*)).

## Taxonomic distribution of DUI in bivalves: an update

Including the two species in the present study, DUI has been reported to date in 104 bivalve species belonging to four subclasses (Heterodonta, Palaeoheterodonta, Pteriomorphia, Protobranchia (but see above comments regarding the questionable inclusion of Nuculanoida within the Protobranchia)), four orders (Mytiloida, Nuculanoida, Unionoida, Veneroida), nine superfamilies (Arcticoidea, Hyrioidea, Mactroidea, Mytiloidea, Nuculanoidea, Solenoidea, Tellinoidea, Unionoidea, Veneroidea) and twelve families (Arcticidae, Donacidae, Hyriidae, Mactridae Margaritiferidae, Mytilidae, Nuculanidae, Semelidae, Solenidae, Unionidae Veneridae, Yoldiidae) (Table 4 and Figs. 1 –4). However, DUI is certainly more widespread in the Bivalvia given that its detection remains difficult; the higher rate of molecular evolution of M type mitochondrial genomes may make it less likely that "universal" mitochondrial primers will anneal to and amplify the M

type (*Theologidis et al., 2008*; *Zouros, 2013*). In addition, the process of mitochondrial genome "masculinization," i.e., when an F genome invades the male route of transmission, can also make the paternally-transmitted genome almost indistinguishable from the maternally-transmitted one (*Stewart et al., 2009*; *Theologidis et al., 2008*; *Zouros, 2013*). Additional studies of bivalves and other mollusc species will significantly contribute to better understanding the taxonomic distribution of the system of doubly uniparental inheritance of mtDNA.

## Phylogenetic analyses and the origin of DUI

As for other veneroid and nuculanoid species (e.g., *Theologidis et al., 2008*), the sex-linked mtDNA sequences of *S. plana* and *Y. hyperborean*, respectively, exhibit a phylogenetic pattern in which the F and M mtDNA sequences are different from one another but yet cluster together in a monophyletic group (Figs. 1–4). Such a pattern can also be seen in the order Mytiloida (Figs. 1–4), except for the Mytilus species complex, which is in agreement with previous studies (e.g., *Doucet-Beaupré et al., 2010*). On the other hand, the observed F/M phylogeny of unionoids contrasts with the patterns observed in the mytiloids, nuculanoids and veneroids. In unionoids, all of the F sequences cluster together and all of the M genomes cluster together such that the F sequences form a monophyletic clade and the M sequences form a monophyletic clade. Similar results have previously been obtained and have suggested that the M and F lineages in the order Unionoida have been distinct and maintained for at least 200 million years (e.g., *Doucet-Beaupré et al., 2010*). The observed phylogenetic pattern of the unionoids could be related to the absence of a masculinization event in this group for over 200 million years (*Hoeh, Stewart & Guttman, 2002*). One hypothesis explaining why F-to-M masculinization events do not occur in freshwater mussels involves the *cox2* extension present only in the M genome (*Curole & Kocher, 2002*). If this extension is essential for the function or transmission of the M genome (*Curole & Kocher, 2002*; *Chakrabarti et al., 2006*; *Chakrabarti et al., 2007*) then masculinization events are no longer possible in this group, unless F genomes are able to obtain, by recombination, those male-specific elements necessary for being paternally transmitted (*Stewart et al., 2009*; *Doucet-Beaupré et al., 2010*). The M genome is considered as a "nearly selfish" element in the sense that it does fulfill its function only if this work cannot be done by the F genome. In other words, paternally-transmitted mt genomes only perform male specific functions. This restricted functionality of the M mtDNA to male gonad tissue is one factor that may explain why the M genome usually evolves faster than the F genome in species with DUI, i.e., because of relaxed selective constraints (*Stewart et al., 1996*; *Zouros, 2000*; *Passamonti, Boore & Scali, 2003*). The other hypotheses that have been proposed to explain the higher rate of evolution of the M genome include (i) a higher number of male germ line mitotic divisions preceding gametogenesis compared to the female germ line (*Selwood, 1968*; *Zwaan & Mathieu, 1992*; *Stewart et al., 1996*), (ii) a greater degree of oxidative damage in metabolically active sperm relative to eggs (*Stewart et al., 1996*; *Zouros, 2013*) and/or (iii) a smaller effective population size of male mitochondria compared to female mitochondria (*Stewart et al., 1996*; *Zouros, 2013*).

As for other recent morphological and molecular studies based on mtDNA and nuclear sequences (*Giribet & Wheeler, 2002*; *Giribet & Distel, 2003*; *Bieler & Mikkelsen, 2006*; *Plazzi & Passamonti, 2010*; *Plazzi et al., 2011*; S Breton et al., 2016, unpublished data), our results support a more derived Nuculanoida clustering with Pteriomorphia instead of a basal position of Nuculanoida + Opponobranchia (=Protobranchia). Therefore, the question about the origin of DUI in the branch leading to the Autolamellibranchia about 460 Mya (*Little & Vrijenhoek, 2003*) or much earlier, perhaps in the early Cambrian (*Boyle & Etter, 2013*), still remains open. The taxonomic position of *Y. hyperborea* and other members of the order Nuculanoida should be tested using an expanded data set. A robust bivalve phylogeny, as well as a much more accurate understanding of the taxonomic distribution of DUI, is needed to allow a rigorous evaluation of a single vs. multiple origins of DUI.

Figures 1–4 demonstrate one split between M and F types at the base of the Unionoida, and one split for each species exhibiting a taxon-specific pattern. Taken at face value, this pattern implies a large number of independent origins of DUI. Given the rare molecular and developmental complexity associated to the DUI system, this hypothesis seems unlikely but cannot be completely rejected yet (*Zouros, 2013*). The opposite hypothesis of a single origin of DUI can only be true if associated with masculinization events along each branch of the phylogenetic tree of the Bivalvia where F and M types are each other's closest relatives. Such masculinization events have already been clearly demonstrated in Mytilus (reviewed in *Stewart et al., 2009*; *Zouros, 2013*), and evidence suggests that mitochondrial recombination and acquisition of key elements of the evolutionarily older M mt genome (i.e., sperm transmission elements) are necessary for a F genome to be transmitted via sperm (e.g., *Stewart et al., 2009*; *Zouros, 2013*; *Kyriakou et al., 2015*). Because masculinization events restore nucleotide divergence between F and M mtDNAs to zero for most of the genes in each of the sex-associated genome (except for key sperm transmission elements), this phenomenon could explain the F/M phylogenetic patterns of mytiloids, nuculanoids and veneroids. As a consequence of the many similarities found among the distantly-related DUI species (e.g., sex ratio bias, mitochondria's behavior in the newly formed zygotes, rates of evolution of the two genomes), and because of the complexity of the DUI system, one could thus favor the hypothesis of a single origin of DUI with repeated masculinization events. However, recent *in silico* analyses of the novel mtDNA-encoded protein-coding genes, i.e., ORFan genes with no known homologous, which have been found in species with DUI support either a viral or a mitochondrial origin for these genes that are most probably involved in the DUI mechanism (*Milani et al., 2013*; *Milani et al., 2014*; *Mitchell et al., 2016*; *Milani, Ghiselli & Passamonti, in press*). These results suggest the possibility of DUI systems with elements of different sources/origins and different mechanisms of action in the distantly-related DUI taxa (i.e., DUI could be achieved by different modifications of strictly maternal inheritance of mitochondria), and this scenario would best fit multiple origins of DUI and the necessary factors.

To conclude, our study presents evidence for the existence of DUI in the nuculanoid species *Yoldia hyperborea* and the veneroid species *Scrobicularia plana*. Because the taxonomic position of *Y. hyperborea* and its order Nuculanoida has been debated over the years and still remain uncertain, the question about the origin of DUI during the Cambrian

or before is still unresolved. A much more accurate understanding of the taxonomic distribution of DUI across the Bivalvia appears to be a priority to help confirming the single or multiple origins of this unusual system of mitochondrial heredity. Testing for its presence in all bivalve superfamilies, especially the one not studied yet (e.g., Solemyoidea, Lucinoidea, Carditoidea), is the only way to make a definitive statement.

## ACKNOWLEDGEMENTS

We would like to thank Cindy Grant and Françoise Denis for providing us the samples, El-Amine Mimouni for his assistance with the phylogenetic analyses, and two anonymous reviewers for insightful criticisms and suggestions.

### Funding

This work was supported by funding from the Natural Sciences and Engineering Research Council of Canada (grant no., RGPIN/435656-2013 to S.B. and grant no., RGPIN/217175-2013 to D.T.S.), and by 'Canziani Bequest' and 'Fondazione del Monte' funding (M.P.). A.G. was financially supported by the Groupe de Recherche Interuniversitaire en Limnologie et en environnement aquatique (GRIL) and by the Faculté des Études Supérieures et Postdoctorales (FESP) of the University of Montreal. The funders had no role in study design, data collection and analysis, decision to publish, or preparation of the manuscript.

### Grant Disclosures

The following grant information was disclosed by the authors:
Natural Sciences and Engineering Research Council of Canada: RGPIN/435656-2013, RGPIN/217175-2013.
'Canziani Bequest' and 'Fondazione del Monte'.
Groupe de Recherche Interuniversitaire en Limnologie et en environnement aquatique (GRIL).
Faculté des Études Supérieures et Postdoctorales (FESP) of the University of Montreal.

### Competing Interests

The authors declare there are no competing interests.

### Author Contributions

- Arthur Gusman performed the experiments, analyzed the data, wrote the paper, prepared figures and/or tables.
- Sophia Lecomte performed the experiments.
- Donald T. Stewart and Marco Passamonti reviewed drafts of the paper.
- Sophie Breton conceived and designed the experiments, contributed reagents/materials/analysis tools, wrote the paper, prepared figures and/or tables, reviewed drafts of the paper.

### DNA Deposition

The following information was supplied regarding the deposition of DNA sequences:

GenBank accession numbers: KX447420, KX447421, KX447422, KX447423, KX447424, KX447425, KX447426, KX447427, KX447428.

### Data Availability

Breton, Sophie (2016): Phylogenetic Dataset Nucleotide.fas. figshare. https://dx.doi.org/10.6084/m9.figshare.3798789.v1.

### Supplemental Information

Supplemental information for this article can be found online at http://dx.doi.org/10.7717/peerj.2760#supplemental-information.

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
