# Peer review of "Pursuing the quest for better understanding the taxonomic distribution of the system of doubly uniparental inheritance of mtDNA"

_PeerJ, doi:10.7717/peerj.2760_

## Round 0.1 · original submission · Minor Revisions

Two reviewers have assessed your paper and both agree that is potentially of interest to readers of the journal.

They both have a few issues that would need to be satisfactorily addressed however.

In particular, reviewer 1 suggests that you should consider removing the two species with very low sample sizes. I agree with this suggestion. It would make the paper neater.

In addition, the minor suggestions of both reviewers should be addressed.

Reviewer 1 ·

Basic reporting

No comments

Experimental design

1. They have few samples and few sequence data to include Cerastoderma edule and Musculus discors in this paper. They only have a few sequences from one gene for one species male and 2 genes for the other species male, which they used universal primers to obtain. Their argument these species do not have DUI would contribute more to the discussion of the taxonomic distribution of the phenomena in Bivalvia with more thorough data as such they don’t really include these species in their analysis or discussion. The paper would be cleaner if they omitted these species from the paper.

Validity of the findings

2. Line 111 capitalize August
3. Line 149 given cloning with pgem only samples one strand of PCR product and taq error is notorious did they send replicate clones for sequencing?
4. If the 2 species with little evidence of DUI were removed from Table 2 there would be room to include table S1 or S2 in the main paper.
5. Line 294 more robust phylogenies have shown monophyly of the Protobranchs please see
-Andrade, et al. 2014. A phylogenetic backbone for Bivalvia: An RNA seq approach. Proc Roy Soc B 282.20142332
- Sharma et al. 2013 Into the deep: A phylogenetic approach to the bivalve subclass Protobranchia. Molecular Phylogenetics and Evolution.
Phylogenies based on single mitochondrial genes for the Bivalvia tend to be problematic.

Reviewer 2 ·

Basic reporting

This manuscript by Gusman and co-authors is an important, albeit incremental, contribution to our understanding of the evolution of the fascinating system of mtDNA inheritance in bivalve molluscs.

I only have a few very minor comments
Line 62: insert comma after "et al." in both citations
Line 80: insert "freshwater" before "bivalve order Unionioida"
Line 80: Typo in Margaritiferidae
Line 111: Capitalize "August"
Lines 214-15: Suggest reporting on a p-value of zero differently here

Experimental design

The methods appear to be appropriate.

Validity of the findings

The interpretations appear to be correct. The findings are somewhat incremental in nature and have not really resulted in any new understanding of DUI and thus mostly confirmatory in nature. I assume that the co-authors are in the process of acquiring samples the remaining bivalve superfamilies in order to complete the picture.

---

## Round 0.2 · accepted · Accept

The changes suggested have been made and the paper is now acceptable for publication.